# Structural Priming Demonstrates Abstract Grammatical Representations in Multilingual Language Models

**James A. Michaelov**[a]*    **Catherine Arnett**[b]*    **Tyler A. Chang**[a]    **Benjamin K. Bergen**[a]

[a]Department of Cognitive Science,
[b]Department of Linguistics,
University of California San Diego
{j1michae, ccarnett, tachang, bkbergen}@ucsd.edu

## Abstract

Abstract grammatical knowledge—of parts of speech and grammatical patterns—is key to the capacity for linguistic generalization in humans. But how abstract is grammatical knowledge in large language models? In the human literature, compelling evidence for grammatical abstraction comes from structural priming. A sentence that shares the same grammatical structure as a preceding sentence is processed and produced more readily. Because confounds exist when using stimuli in a single language, evidence of abstraction is even more compelling from crosslingual structural priming, where use of a syntactic structure in one language primes an analogous structure in another language. We measure crosslingual structural priming in large language models, comparing model behavior to human experimental results from eight crosslingual experiments covering six languages, and four monolingual structural priming experiments in three non-English languages. We find evidence for abstract monolingual and crosslingual grammatical representations in the models that function similarly to those found in humans. These results demonstrate that grammatical representations in multilingual language models are not only similar across languages, but they can causally influence text produced in different languages.

## 1 Introduction

What do language models learn about the structure of the languages they are trained on? Under both more traditional generative (Chomsky, 1965) and cognitively-inspired usage-based theories of language (Tomasello, 2003; Goldberg, 2006; Bybee, 2010), the key to generalizable natural language comprehension and production is the acquisition of grammatical structures that are sufficiently abstract to account for the full range of possible sentences in a language. In fact, both theoretical and experimental accounts of language suggest that grammatical

representations are abstract enough to be shared across languages in both humans (Heydel and Murray, 2000; Hartsuiker et al., 2004; Schoonbaert et al., 2007) and language models (Conneau et al., 2020b,a; Jones et al., 2021).

The strongest evidence for grammatical abstraction in humans comes from *structural priming*, a widely used and robust experimental paradigm. Structural priming is based on the hypothesis that grammatical structures may be activated during language processing. Priming then increases the likelihood of production or increased ease of processing of future sentences sharing the same grammatical structures (Bock, 1986; Ferreira and Bock, 2006; Pickering and Ferreira, 2008; Dell and Ferreira, 2016; Mahowald et al., 2016; Branigan and Pickering, 2017). For example, Bock (1986) finds that people are more likely to produce an active sentence (e.g. *one of the fans punched the referee*) than a passive sentence (e.g. *the referee was punched by one of the fans*) after another active sentence. This has been argued (Bock, 1986; Heydel and Murray, 2000; Pickering and Ferreira, 2008; Reitter et al., 2011; Mahowald et al., 2016; Branigan and Pickering, 2017) to demonstrate common abstractions generalized across all sentences with the same structure, regardless of content.

Researchers have found evidence that structural priming for sentences with the same structure occurs even when the two sentences are in different languages (Loebell and Bock, 2003; Hartsuiker et al., 2004; Schoonbaert et al., 2007; Shin and Christianson, 2009; Bernolet et al., 2013; van Gompel and Arai, 2018; Kotzochampou and Chondrogianni, 2022). This *crosslingual structural priming* takes abstraction one step further. First, it avoids any possible confounding effects of lexical repetition and lexical priming of individual words—within a given language, sentences with the same structure often share function words (for discussion, see Sinclair et al., 2022). More fundamentally,

---
*Equal contribution.

crosslingual structural priming represents an extra degree of grammatical abstraction not just within a language, but across languages.

We apply this same logic to language models in the present study. While several previous studies have explored structural priming in language models (Prasad et al., 2019; Sinclair et al., 2022; Frank, 2021; Li et al., 2022; Choi and Park, 2022), to the best of our knowledge, this is the first to look at crosslingual structural priming in Transformer language models. We replicate eight human psycholinguistic studies, investigating structural priming in English, Dutch (Schoonbaert et al., 2007; Bernolet et al., 2013), Spanish (Hartsuiker et al., 2004), German (Loebell and Bock, 2003), Greek (Kotzochampou and Chondrogianni, 2022), Polish (Fleischer et al., 2012), and Mandarin (Cai et al., 2012). We find priming effects in the majority of the crosslingual studies and all of the monolingual studies, which we argue supports the claim that multilingual models have shared grammatical representations across languages that play a functional role in language generation.

## 2 Background

Structural priming effects have been observed in humans both within a given language (Bock, 1986; Ferreira and Bock, 2006; Pickering and Ferreira, 2008; Dell and Ferreira, 2016; Mahowald et al., 2016; Branigan and Pickering, 2017) and crosslingually (Loebell and Bock, 2003; Hartsuiker et al., 2004; Schoonbaert et al., 2007; Shin and Christianson, 2009; Bernolet et al., 2013; van Gompel and Arai, 2018; Kotzochampou and Chondrogianni, 2022). In language models, previous work has demonstrated structural priming effects in English (Prasad et al., 2019; Sinclair et al., 2022; Choi and Park, 2022), and initial results have found priming effects between English and Dutch in LSTM language models (Frank, 2021). As these studies argue, the structural priming approach avoids several possible assumptions and confounds found in previous work investigating abstraction in grammatical learning. For example, differences in language model probabilities for individual grammatical vs. ungrammatical sentences may not imply that the models have formed abstract grammatical representations that generalize across sentences (Sinclair et al., 2022); other approaches involving probing (e.g. Hewitt and Manning, 2019; Chi et al., 2020) often do not test whether the internal model states

are causally involved in the text predicted or generated by the model (Voita and Titov, 2020; Sinclair et al., 2022). The structural priming paradigm allows researchers to evaluate whether grammatical representations generalize across sentences in language models, and whether these representations causally influence model-generated text. Furthermore, structural priming is agnostic to the specific language model architecture and does not rely on direct access to internal model states.

However, the structural priming paradigm has not been applied to modern multilingual language models. Previous work has demonstrated that multilingual language models encode grammatical features in shared subspaces across languages (Chi et al., 2020; Chang et al., 2022; de Varda and Marelli, 2023), largely relying on probing methods that do not establish causal effects on model predictions. Crosslingual structural priming would provide evidence that the abstract grammatical representations shared across languages in the models have causal effects on model-generated text. It would also afford a comparison between grammatical representations in multilingual language models and human bilinguals. These shared grammatical representations may help explain crosslingual transfer abilities in multilingual models, where tasks learned in one language can be transferred to another (Artetxe et al., 2020; Conneau et al., 2020a,b; K et al., 2020; Goyal et al., 2021; Ogueji et al., 2021; Armengol-Estapé et al., 2021, 2022; Blevins and Zettlemoyer, 2022; Chai et al., 2022; Muennighoff et al., 2023; Wu et al., 2022; Guarasci et al., 2022; Eronen et al., 2023).

Thus, this study presents what is to our knowledge the first experiment testing for crosslingual structural priming in Transformer language models. The findings broadly replicate human structural priming results: higher probabilities for sentences that share grammatical structure with prime sentences both within and across languages.

## 3 Method

We test multilingual language models for structural priming using the stimuli from eight crosslingual and four monolingual priming studies in humans. Individual studies are described in §4.

### 3.1 Materials

All replicated studies have open access stimuli with prime sentences for different constructions (§3.3).

Where target sentences are not provided (because participant responses were manually coded by the experimenters), we reconstruct target sentences and verify them with native speakers.

## 3.2 Language Models

We test structural priming in XGLM 4.5B (Lin et al., 2022), a multilingual autoregressive Transformer trained on data from all languages we study in this paper, namely, English, Dutch, Spanish, German, Greek, Polish, and Mandarin. To the best of our knowledge, this is the only available pretrained (and not fine-tuned) autoregressive language model trained on all the aforementioned languages. To avoid drawing any conclusions based on the idiosyncrasies of a single language model, we also test a number of other multilingual language models trained on most of these languages, namely the other XGLM models, i.e., 564M, 1.7B, 2.9B, and 7.5B, which are trained on all the languages except for Dutch and Polish; and PolyLM 1.7B and 13B (Wei et al., 2023), which are trained on all the languages except for Greek.

## 3.3 Grammatical Alternations Tested

We focus on structural priming for the three alternations primarily used in existing human studies.

**Dative Alternation (DO/PO)** Some languages permit multiple orders of the direct and indirect objects in sentences. In PO (prepositional object) constructions, e.g., *the chef gives a hat to the swimmer* (Schoonbaert et al., 2007), the direct object *a hat* immediately follows the verb and the indirect object is introduced with the prepositional phrase *to the swimmer*. In DO (double object) constructions, e.g., *the chef gives the swimmer a hat*, the indirect object *the swimmer* appears before the direct object *a hat* and neither is introduced by a preposition. Researchers compare the proportion of DO or PO sentences produced by experimental participants following a DO or PO prime.

**Active/Passive** In active sentences the syntactic subject is the agent of the action, while in passive sentences the syntactic subject is the patient or theme of the action. E.g., *the taxi chases the truck* is active, and *the truck is chased by the taxi* is passive (Hartsuiker et al., 2004). Researchers compare the proportion of active or passive sentences produced by experimental participants following an active or passive prime.

**Of-/S-Genitive** *Of*- and *S*-Genitives represent two different ways of expressing possessive meaning. In an *of*-genitive, the possessed thing is followed by a preposition such as *of* and then the possessor, e.g., *the scarf of the boy is yellow*. In *s*-genitives in the languages we analyze (English and Dutch), the possessor is followed by a word or an attached morpheme such as *'s* which is then followed by the possessed thing, e.g., *the boy's scarf is yellow* (Bernolet et al., 2013). Researchers compare the proportion of *of*-genitive or *s*-genitive sentences produced by experimental participants following an *of*-genitive or *s*-genitive prime.

## 3.4 Testing Structural Priming in Models

In human studies, researchers test for structural priming by comparing the proportion of sentences (targets) of given types produced following primes of different types. Analogously, for each experimental item, we prompt the language model with the prime sentence and compute the normalized probabilities of each of the two target sentences. We illustrate our approach to computing these normalized probabilities below.

First, consider the example dative alternation stimulus sentences from Schoonbaert et al. (2007):

(1) (a) **DO prime:** The cowboy shows the pirate an apple.
   (b) **PO prime:** The cowboy shows an apple to the pirate.
   (c) **DO target:** The chef gives the swimmer a hat.
   (d) **PO target:** The chef gives a hat to the swimmer.

We can use language models to calculate the probability of each target following each prime by taking the product of the conditional probabilities of all tokens in the target sentence given the prime sentence and all preceding tokens in the target sentence. In practice, these probabilities are very small, but for illustrative purposes, we can imagine these have the probabilities in (2).

(2) (a) P(PO Target | DO Prime) = 0.03
   (b) P(DO Target | DO Prime) = 0.02
   (c) P(PO Target | PO Prime) = 0.04
   (d) P(DO Target | PO Prime) = 0.01

We then normalize these probabilities by calculating the conditional probability of each target sentence given that the model response is one of the two target sentences, as shown in (3).

(3)  (a)  $P_N(PO \mid DO) = 0.03/(0.03+0.02) = 0.60$
  (b)  $P_N(DO \mid DO) = 0.02/(0.03+0.02) = 0.40$
  (c)  $P_N(PO \mid PO) = 0.04/(0.04+0.01) = 0.80$
  (d)  $P_N(DO \mid PO) = 0.01/(0.04+0.01) = 0.20$

Because the normalized probabilities of the two targets following a given prime sum to one, we only consider the probabilities for one target type in our analyses (comparing over the two different prime types). For example, to test for a priming effect, we could either compare the difference between $P_N(PO \mid PO)$ and $P_N(PO \mid DO)$ or the difference between $P_N(DO \mid PO)$ and $P_N(DO \mid DO)$. We follow the original human studies in the choice of which target construction to plot and test.

We run statistical analyses, testing whether effects are significant for each language model on each set of stimuli. To do this, we construct a linear mixed-effects model predicting the target sentence probability (e.g. probability of a PO sentence) for each item. We include a random intercept for experimental item, and we test whether prime type (e.g. DO vs. PO) significantly predicts target structure probability. All reported $p$-values are corrected for multiple comparisons by controlling for false discovery rate (Benjamini and Hochberg, 1995). All stimuli, data, code, and statistical analyses are provided at `https://osf.io/2vjw6/`.

## 4 Results

In reporting whether the structural priming effects from human experiments replicate in XGLM language models, we primarily consider the direction of each effect in the language models (e.g. whether PO constructions are more likely after PO vs. DO primes) rather than effect sizes or raw probabilities. The mean of the relative probabilities assigned by language models to the different constructions in each condition may not be directly comparable to human probabilities of production. Humans are sensitive to contextual cues that may not be available to language models; notably, in these tasks, humans are presented with pictures corresponding to events in the structural priming paradigm. Furthermore, construction probabilities in language models may be biased by the frequency of related constructions in any of the many languages on which the models are trained. Thus, we focus only on whether the language models replicate the direction of the principal effect in each human study.

## 4.1 Crosslingual Structural Priming

We test whether eight human crosslingual structural priming studies replicate in language models. These studies cover structural priming between English and Dutch (Schoonbaert et al., 2007; Bernolet et al., 2013), Spanish (Hartsuiker et al., 2004), German (Loebell and Bock, 2003), Greek (Kotzochampou and Chondrogianni, 2022), and Polish (Fleischer et al., 2012). For each experiment, we show the original human probabilities and the normalized probabilities calculated using each language model, as well as whether there is a significant priming effect (Figure 1). The full statistical results are reported in Appendix B.

### 4.1.1 Schoonbaert et al. (2007): Dutch→English

Schoonbaert et al. (2007) prime 32 Dutch-English bilinguals with 192 Dutch sentences with either prepositional (PO) or dative object (DO) constructions. Schoonbaert et al. (2007) find that experimental participants produce more PO sentences when primed with a PO sentence than when primed with a DO sentence (see Figure 1A). We see the same pattern with nearly all the language models (Figure 1A). With the exception of XGLM 1.7B, where the effect is only marginally significant after correction for multiple comparisons, all language models predict English PO targets to be significantly more likely when they follow Dutch PO primes than when they follow Dutch DO primes.

### 4.1.2 Schoonbaert et al. (2007): English→Dutch

Schoonbaert et al. (2007) also observe DO/PO structural priming from English to Dutch (32 participants; 192 primes). As seen in Figure 1B, all language models show a significant priming effect.

### 4.1.3 Bernolet et al. (2013): Dutch→English

Bernolet et al. (2013) conduct a Dutch→English structural priming experiment with 24 Dutch-English bilinguals on 192 prime sentences, and they find that the production of *s*-genitives is significantly more likely after an *s*-genitive prime than after an *of*-genitive prime. We also observe this in all of the language models, as seen in Figure 1C.

### 4.1.4 Hartsuiker et al. (2004): Spanish→English

Hartsuiker et al. (2004) investigate Spanish→English structural priming with

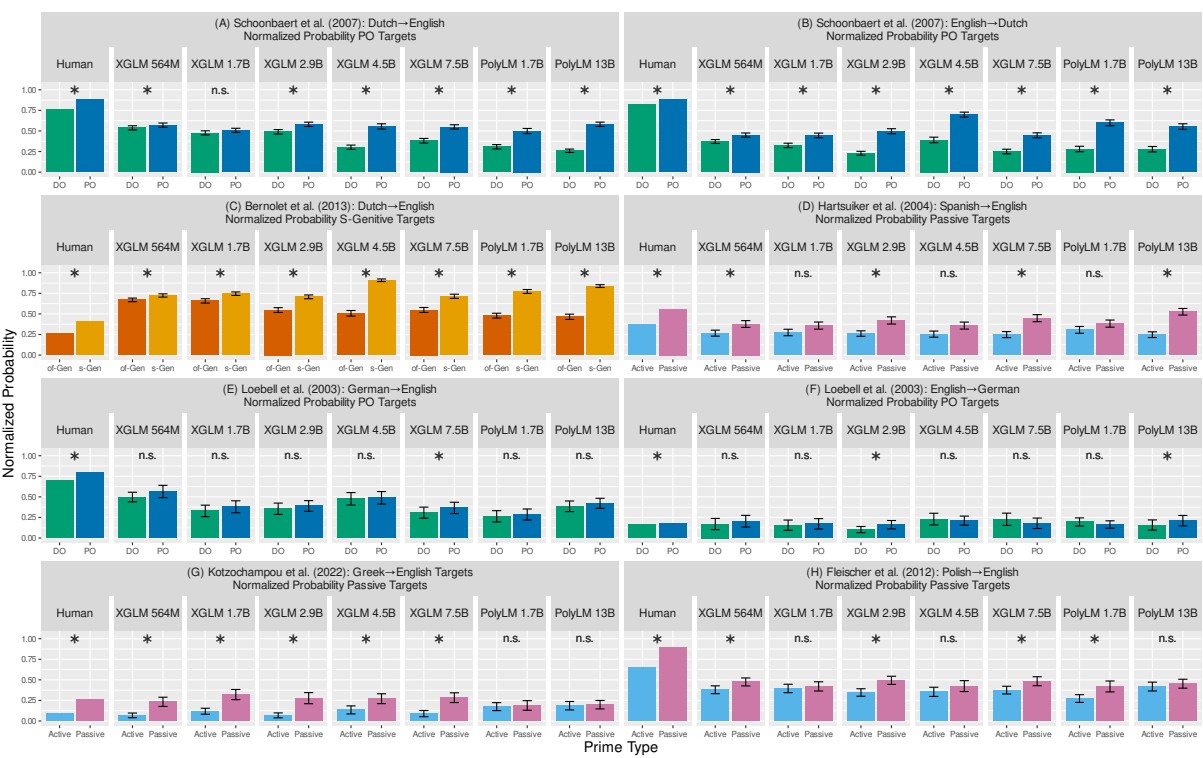

Figure 1: Human and language model results for crosslingual structural priming experiments.

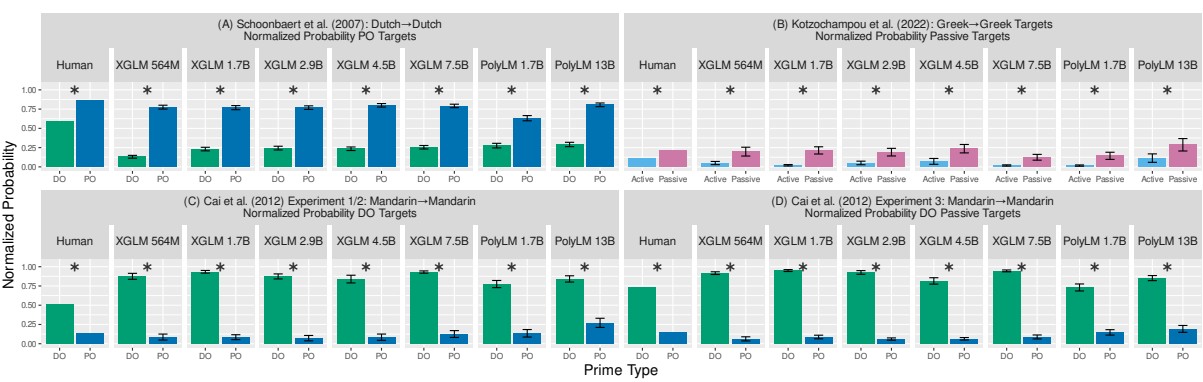

Figure 2: Human and language model results for within-language structural priming experiments.

24 Spanish-English bilinguals on 128 prime sentences, finding a significantly higher proportion of passive responses after passive primes than active primes. As shown in Figure 1D, this effect is replicated by XGLM 564M, 2.9B, and 7.5B as well as PolyLM 13B, with XGLM 4.5B showing a marginal effect ($p = 0.0565$).

### 4.1.5 Loebell and Bock (2003): German→English

Loebell and Bock (2003) find a small but significant priming effect of dative alternation (DO/PO) from German to English with 48 German-English bilinguals on 32 prime sentences. As can be seen in Figure 1E, while all language models show a

numerical effect in the correct direction, the effect is only significant for XGLM 7.5B.

### 4.1.6 Loebell and Bock (2003): English→German

Loebell and Bock (2003) also test 48 German-English bilinguals for a dative alternation (DO/PO) priming effect from English primes to German targets (32 prime sentences), finding a small but significant priming effect. As we show in Figure 1F, the models are relatively varied in direction of numerical difference. However, only XGLM 2.9B and PolyLM 13B display a significant effect, and in both cases the effect is in the same direction as that found with human participants.

### 4.1.7 Kotzochampou and Chondrogianni (2022): Greek→English

Kotzochampou and Chondrogianni (2022) find active/passive priming from Greek to English in 25 Greek-English bilinguals. Participants are more likely to produce passive responses after passive primes (48 prime sentences). As shown in Figure 1G), all XGLMs display this effect, while the PolyLMs, which are not trained on Greek, do not.

### 4.1.8 Fleischer et al. (2012): Polish→English

Similarly, Fleischer et al. (2012) find active/passive priming from Polish to English in 24 Polish-English bilinguals on 64 prime sentences. As we see in Figure 1H, while all models show a numerical difference in the correct direction, the effect is only significant for XGLM 564M, 2.9B, and 7.5B, and for PolyLM 1.7B.

### 4.2 Monolingual Structural Priming

In the previous section, we found crosslingual priming effects in language models for the majority of crosslingual priming studies in humans. However, six of the eight studies have English target sentences. Our results up to this point primarily show an effect of structural priming on English targets. While both previous work (Sinclair et al., 2022) and our results in §4.1 may indeed demonstrate the effects of abstract grammatical representations on generated text in English, we should not assume that such effects can reliably be observed for other languages. Thus, we test whether multilingual language models exhibit within-language structural priming effects comparable to those found in human studies for Dutch (Schoonbaert et al., 2007), Greek (Kotzochampou and Chondrogianni, 2022), and two studies in Mandarin (Cai et al., 2012).

### 4.2.1 Schoonbaert et al. (2007): Dutch→Dutch

Using Dutch prime and target sentences (192 primes), Schoonbaert et al. (2007) find that Dutch-English bilinguals (N=32) produce PO sentences at a higher rate when primed by a PO sentence compared to a DO sentence. As we see in Figure 2A, all language models display this effect.

### 4.2.2 Kotzochampou and Chondrogianni (2022): Greek→Greek

In their Greek→Greek priming experiment, Kotzochampou and Chondrogianni (2022) find an active/passive priming effect in native Greek speakers (N=25) using 48 primes. As shown in Figure 2B, this effect is replicated by all language models.

### 4.2.3 Cai et al. (2012): Mandarin→Mandarin

Using two separate sets of stimuli, Cai et al. (2012) find within-language DO/PO priming effects in native Mandarin speakers (N=28, N=24).[1] As seen in Figure 2C and 2D, all language models show significant effects for both sets of stimuli (48 prime sentences in their Experiments 1 and 2, and 68 prime sentences in their Experiment 3).

### 4.3 Further Tests of Structural Priming

We have now observed within-language structural priming in multilingual language models for languages other than English. In §4.1, we found robust English→Dutch structural priming (Schoonbaert et al., 2007) but only limited priming effects for targets in German. Although there are no human results for the non-English targets in the other studies in §4.1, we can still evaluate crosslingual structural priming with non-English targets in the language models by switching the prime and target sentences in the stimuli. Specifically, we test structural priming from English to Dutch (Bernolet et al., 2013), Spanish (Hartsuiker et al., 2004), Polish (Fleischer et al., 2012), and Greek (Kotzochampou and Chondrogianni, 2022).

All models show a significant effect on the reversed Bernolet et al. (2013) stimuli (Figure 3A; English→Dutch), and all models but PolyLM 1.7B show the same for the reversed Hartsuiker et al. (2004) stimuli (Figure 3B; English→Spanish). The other results are less clear-cut. While XGLM 564M, 2.9B, and 4.5B and the PolyLMs show a numerical effect in the correct direction for the reversed Fleischer et al. (2012) stimuli (English→Polish; Figure 3C), only PolyLM 1.7B shows a significant effect. For the reversed Kotzochampou and Chondrogianni (2022) stimuli (English→Greek; Figure 3D), all the XGLMs and PolyLM 13B show a numerical tendency in the correct direction, but only XGLM 564M and 4.5B show a significant effect.

---

[1]The original study tests the effect of variants of DO/PO primes (topicalized DO/PO and *Ba*-DO; see Cai et al., 2012). To unify our analyses across studies, we only look at structural priming following the canonical DO and PO primes used in both Experiments 1 and 2 of the original study, as well as those used in Experiment 3.

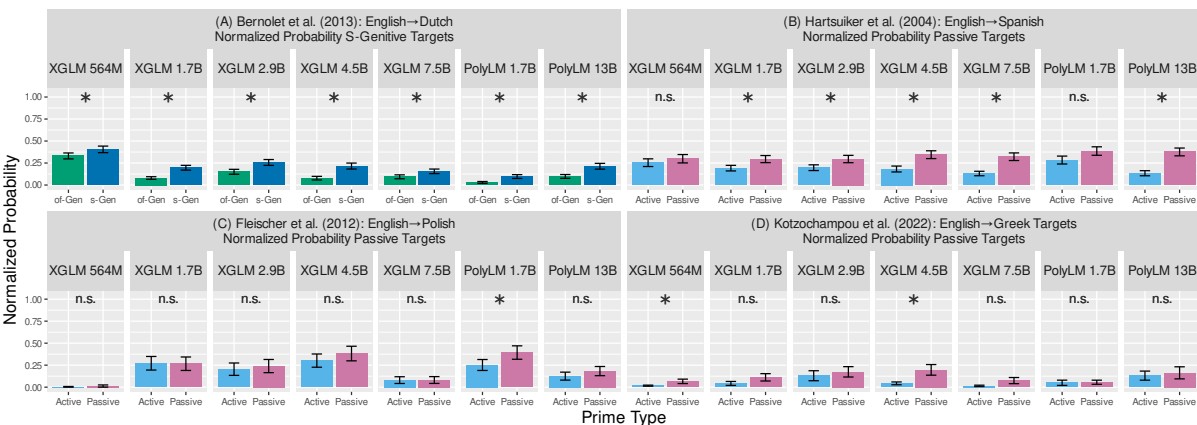

Figure 3: Language model results for structural priming experiments with no human baseline.

## 5 Discussion

We find structural priming effects in at least one language model on each set of stimuli (correcting for multiple comparisons). Moreover, we observe a significant effect in all models with the monolingual stimuli, and in the majority of the models for 8 of the 12 crosslingual stimuli. In line with previous work (Hewitt and Manning, 2019; Chi et al., 2020), this supports the claim that language models learn generalized, abstract, and multilingual representations of grammatical structure. Our results further suggest that these shared grammatical representations are causally linked to model output.

### 5.1 Differences between models

In some ways, we see expected patterns across models. For example, for the XGLMs trained on 30 languages (XGLM 564M, 1.7B, 2.9B, and 7.5B), the larger models tend to display larger effect sizes than the smaller models, in line with the idea that model performance can scale with number of parameters (Brown et al., 2020; Kaplan et al., 2020; Rae et al., 2022; Hoffmann et al., 2022; Touvron et al., 2023). Additionally, the PolyLMs, which are not trained on Greek, do not show crosslingual structural priming for Greek (neither Greek→English nor English→Greek).

On the other hand, one surprising finding is that despite not being trained on Greek, the PolyLMs are able to successfully model monolingual structural priming in Greek. The most likely explanation for this is what Sinclair et al. (2022) refer to as 'lexical overlap'—the overlap of function words between primes and targets substantially boosts structural priming effects. In the

same way that humans find it easier to process words that have recently been mentioned (Rugg, 1985, 1990; Van Petten et al., 1991; Besson et al., 1992; Mitchell et al., 1993; Rommers and Federmeier, 2018), language models may predict that previously-mentioned words are more likely to occur again (a familiar phenomenon in the case of repeated text loops; see Holtzman et al., 2020; See et al., 2019; Xu et al., 2022) even if they are not trained on the words explicitly. This would explain the results for the Kotzochampou and Chondrogianni (2022) stimuli, as the Greek passive stimuli always include the word από.

Such an explanation could also account for the performance of XGLM 564M, 1.7B, 2.9B, and 7.5B on the Dutch and Polish stimuli. Despite not being intentionally trained on Dutch or Polish, we see robust crosslingual Dutch→English and English→Dutch structural priming, as well as Polish→English structural priming, in three of these models. However, as discussed previously, crosslingual structural priming avoids the possible confound of lexical overlap. For these results, therefore, a more likely explanation is language contamination. In contemporaneous work, we find that training on fewer than 1M tokens in a second language is sufficient for structural priming effects to emerge (Arnett et al., 2023); our estimates of the amount of language contamination in XGLM 564M, 1.7B, 2.9B, and 7.5B range from 1.77M tokens of Dutch and 1.46M tokens of Polish at the most conservative to 152.5M and 33.4M tokens respectively at the most lenient (see Appendix A).

The smaller amount of Polish contamination, as well as the fact that Polish is less closely related to English, may explain the less consistent

Polish→English structural priming effects and the virtually non-existent English→Polish effects in these models, but as will be discussed in §5.2, there may be other reasons for this latter pattern.

## 5.2 Null Effects and Asymmetries

More theoretically interesting is the question of why some language models fail to display crosslingual structural priming on some sets of stimuli, even when trained on both languages. For example, in the Loebell and Bock (2003) replications, only XGLM 7.5B shows a significant effect of German→English structural priming, and only XGLM 2.9B and PolyLM 13B show a significant effect of English→German structural priming. This may be due to the grammatical structures used in the stimuli (DO/PO). While the original study does find crosslingual structural priming effects, the effect sizes are small; the authors suggest that this may partly be because "the prepositional form is used more restrictively in German" (Loebell and Bock, 2003, p. 807).

We also see an asymmetry in the crosslingual structural priming effects between some languages. While the effects in the Dutch→English (Bernolet et al., 2013) and Spanish→English (Hartsuiker et al., 2004) studies mostly remain when the direction of the languages is reversed, this is not the case for the Polish→English (Fleischer et al., 2012) and Greek→English (Kotzochampou and Chondrogianni, 2022) results. This may be due to the smaller quantity of training data for Polish and Greek compared to Spanish in XGLM. While XGLM is only trained on slightly more Dutch than Polish, Dutch is also more similar to English in terms of its lexicon and morphosyntax, so it may benefit from more effective crosslingual transfer (Conneau et al., 2020b; Gerz et al., 2018; Guarasci et al., 2022; Winata et al., 2022; Ahuja et al., 2022; Oladipo et al., 2022; Eronen et al., 2023).

If it is indeed the case that structural priming effects in language models are weaker when the target language is less trained on, this would contrast with human studies, where crosslingual structural priming appears most reliable when the prime is in participants' native or primary language (L1) and the target is in their second language (L2). The reverse case often results in smaller effect sizes (Schoonbaert et al., 2007) or effects that are not significant at all (Shin, 2010). Under this account, language models' dependence on target language train-

ing and humans' dependence on prime language experience for structural priming would suggest that there are key differences between the models and humans in how grammatical representations function in comprehension and production.

An alternative reason for the absence of crosslingual structural priming effects for the English→Polish and English→Greek stimuli is a combination of model features and features of the languages themselves. For example, structural priming effects at the syntactic level may overall be stronger for English targets. English is a language with relatively fixed word order, and thus, competence in English may require a more explicit representation of word order than other languages. In contrast to English, Polish and Greek are morphologically rich languages, where important information is conveyed through morphology (e.g. word inflections), and word orders are less fixed (Tzanidaki, 1995; Siewierska, 1993). Thus, structural priming effects with Polish and Greek targets would manifest as differences in target sentence morphology. However, contemporary language models such as XGLM have a limited ability to deal with morphology. Most state-of-the-art models use WordPiece (Wu et al., 2016) or Sentence-Piece (Kudo and Richardson, 2018) tokenizers, but other approaches may be necessary for morphologically rich languages (Klein and Tsarfaty, 2020; Park et al., 2021; Soulos et al., 2021; Nzeyimana and Niyongabo Rubungo, 2022; Seker et al., 2022).

Thus, while humans are able to exhibit crosslingual structural priming effects between languages when the equivalent structures do not share the same word orders (Muylle et al., 2020; Ziegler et al., 2019; Hsieh, 2017; Chen et al., 2013), this may not hold for contemporary language models. Specifically, given the aforementioned limitations of contemporary language models, it would be unsurprising that structural priming effects are weaker for morphologically-rich target languages with relatively free word order such as Polish and Greek.

## 5.3 Implications for Multilingual Models

The results reported here seem to bode well for the crosslingual capacities of multilingual language models. They indicate shared representations of grammatical structure across languages (in line with Chi et al., 2020; Chang et al., 2022; de Varda and Marelli, 2023), and they show that these representations have a causal role in language generation.

The results also demonstrate that crosslinguistic transfer can take place at the level of grammatical structures, not just specific phrases, concepts, and individual examples. Crosslinguistic generalizations can extend at least to grammatical abstractions, and thus learning a grammatical structure in one language may aid in the acquisition of its homologue in a second language.

How do language models acquire these abstractions? As Contreras Kallens et al. (2023) point out, language models learn grammatical knowledge through exposure. To the degree that similar outcomes for models and humans indicate shared mechanisms, this serves to reinforce claims of usage-based (i.e. functional) accounts of language acquisition (Tomasello, 2003; Goldberg, 2006; Bybee, 2010), which argue that statistical, bottom-up learning may be sufficient to account for abstract grammatical knowledge. Specifically, the results of our study demonstrate the in-principle viability of learning the kinds of linguistic structures that are sensitive to structural priming using the statistics of language alone. Indeed, under certain accounts of language (e.g. Branigan and Pickering, 2017), it is precisely the kinds of grammatical structures that can be primed that *are* the abstract linguistic representations that we learn when we acquire language. Our results are thus in line with Contreras Kallens et al.'s (2023) argument that it may be possible to use language models as tests for necessity in theories of grammar learning. Taking this further, future work might use different kinds of language models to test what types of priors or biases, if any, are required for any learner to acquire abstract linguistic knowledge.

In practical terms, the structural priming paradigm is an innovative way to probe whether a language model has formed an abstract representation of a given structure (Sinclair et al., 2022), both within and across languages. By testing whether a structure primes a homologous structure in another language, we can assess whether the model's representation for that structure is abstract enough to generalize beyond individual sentences and has a functional role in text generation. As language models are increasingly used in text generation scenarios (Lin et al., 2022) rather than fine-tuning representations (Conneau et al., 2020a), understanding the effects of such representations on text generation is increasingly important. Previous work has compared language models to human studies of

language comprehension (e.g. Oh and Schuler, 2023; Michaelov et al., 2022; Wilcox et al., 2021; Hollenstein et al., 2021; Kuribayashi et al., 2021; Goodkind and Bicknell, 2018), and while the degree to which the the mechanisms involved in comprehension and production differ in humans is a matter of current debate (Pickering and Garrod, 2007, 2013; Hendriks, 2014; Meyer et al., 2016; Martin et al., 2018), our results show that human studies of language production can also be reproduced in language models used for text generation.

## 6 Conclusion

Using structural priming, we measure changes in probability for target sentences that do or do not share structure with a prime sentence. Analogously to humans, models predict that a similar target structure is generally more likely than a different one, whether within or across languages. We observe several exceptions, which may reveal features of the languages in question, limitations of the models themselves, or interactions between the two. Based on our results, we argue that multilingual autoregressive Transformer language models display evidence of abstract grammatical knowledge both within and across languages. Our results provide evidence that these shared representations are not only latent in multilingual models' representation spaces, but also causally impact their outputs.

## Limitations

To ensure that the stimuli used for the language models indeed elicit structural priming effects in people, we only use stimuli made available by the authors of previously-published studies on structural priming in humans. Thus, our study analyzes only a subset of possible grammatical alternations and languages. All of our crosslingual structural priming stimuli involve English as one of the languages, and all other languages included are, with the exception of Mandarin, Indo-European languages spoken in Europe. All are also moderately or highly-resourced in the NLP literature (Joshi et al., 2020). Thus, our study is not able to account for the full diversity of human language.

Additionally, while psycholinguistic studies often take crosslingual structural priming to indicate shared representations, there are alternate interpretations. Most notably, because structurally similar sentences are more likely to occur in succession than chance, it is possible that increased proba-

bility for same-structure target sentences reflects likely co-occurrence of distinct, associated representations, rather than a single, common, abstract representation (Ahn and Ferreira, 2023). While this is a much more viable explanation for monolingual than crosslingual priming, the presence of even limited code-switching in training data could in principle lead to similar effects across languages.

## Ethics Statement

Our work complies with the ACL Ethics Policy, and we believe that testing how well language models handle languages other than English is an important avenue of research to reduce the potential harms of language model applications. We did not train any models for this study; instead, we used the pre-trained XGLM (Lin et al., 2022) and PolyLM (Wei et al., 2023) families of models made available through the transformers Python package (Wolf et al., 2020). All analyses were run on an NVIDIA RTX A6000 GPU, running for a total of 4 hours.

## Acknowledgements

We would like to thank Sarah Bernolet, Kathryn Bock, Holly P. Branigan, Zhenguang G. Cai, Vasiliki Chondrogianni, Zuzanna Fleischer, Robert J. Hartsuiker, Sotiria Kotzochampou, Helga Loebell, Janet F. McLean, Martin J. Pickering, Sofie Schoonbaert, and Eline Veltkamp for making their experimental stimuli available; and Nikitas Angeletos Chrysaitis, Pamela D. Rivière Ruiz, Stephan Kaufhold, Quirine van Engen, Alexandra Taylor, Robert Slawinski, Felix J. Binder, Johanna Meyer, Tiffany Wu, Fiona Tang, Emily Xu, and Jason Tran for their assistance in preparing them for use in the present study. Models were evaluated using hardware provided by the NVIDIA Corporation as part of an NVIDIA Academic Hardware Grant. Tyler Chang is partially supported by the UCSD HDSI graduate fellowship.

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

## A Language Contamination in Multilingual Language Models

In this section, we estimate language contamination in CC-100-XL, the dataset used to train the XGLM models. While the dataset itself is not made available by Lin et al. (2022), the procedure used for language identification is similar to CC-100 (Conneau et al., 2020a; Wenzek et al., 2020).

While there are some differences in the approaches used for filtering languages to ensure high-quality data, both corpora are based on Common-Crawl snapshots and are divided into languages

| Language ID Tool | Dutch | | Polish | |
| --- | --- | --- | --- | --- |
| | Proportion | Estimated Tokens | Proportion | Estimated tokens |
| cld3 | 0.03051% | 152,528,079 | 0.00668% | 33,418,112 |
| fastText | 0.00212% | 10,595,403 | 0.00157% | 7,841,824 |
| Consensus (cld3 + fastText) | 0.00035% | 1,774,765 | 0.00029% | 1,456,856 |

Table 1: Estimated Dutch and Polish contamination in the training data of XGLM 564M, 1.7B, 2.9B, and 7.5B, based on language identification using cld3 and fastText, only considering tokens that both language identification models predict to be Dutch or Polish.

using the fastText language identification model (Joulin et al., 2017). Both CC-100 and CC-100-XL also involve a further language identification step. For CC-100, an unnamed internal tool is also used for language identification; for CC-100-XL, an additional step of language identification takes place where text language is also identified at the paragraph level.

To test for Dutch and Polish contamination, we sample roughly 100M tokens (based on the XGLM 7.5B tokenizer) of all languages in the replicated CC-100 dataset[2] that XLGM 564M, 1.7B, 2.9B, and 7.5B are trained on. We only consider languages that have 100M or more tokens in CC-100 and that either use the Latin alphabet (Spanish, French, Italian, Portuguese, Finnish, Indonesian, Turkish, Vietnamese, Catalan, Estonian, Swahili, Basque), are Slavic (Russian, Bulgarian), or both (English, German). Specifically, we sample from each of these languages until we have enough documents that the number of tokens in each language is at least 100M. Thus, our sample of CC-100 includes roughly 1.6B tokens.

To replicate the additional filtering of CC-100-XL, we split all documents by paragraph and run language identification on them using the latest version of the fastText language identification model released as part of the "No Language Left Behind" project (Costa-jussà et al., 2022). We set the identification threshold to 0.5, which the authors find to be effective for lower-resource languages (which some of our sampled languages are among). We note that this is a newer and likely more accurate version of the language identification model than that used to create CC-100-XL, and thus it is even less likely to include data from languages other than those intended. We only analyze the data from paragraphs identified to be the same language as the document label.

To identify Dutch and Polish in these paragraphs,

we divide paragraphs into sentences by splitting at each period character, and we run each sentence through both the aforementioned latest version of the fastText language identification model (Costa-jussà et al., 2022; Joulin et al., 2017) and the cld3 language identifier (Xue et al., 2021) as provided in the gcld3 python package (Al-Rfou, 2020). We use a stricter threshold of 0.9 (as recommended for high-resource languages; Costa-jussà et al., 2022) for the former and use the default threshold of 0.7 for the latter.[3]

To estimate the total amount of contamination in each of these languages, we calculate the proportion of each language sample that includes Dutch or Polish. We then multiply this by the number of tokens in each language, which we estimate by multiplying the proportions given in Figure 1 of Lin et al. (2022) by 500B, the total number of tokens. We first provide two estimates of contamination for Dutch and Polish in Table 1: the amount of contamination as identified by the fastText language identification model, and the amount identified by cld3. We also provide a third, more conservative estimate, that only includes the tokens that both language identification models identify as either Dutch or Polish. We note that because we only look at data from 16 of the 30 training languages, these numbers are likely to substantially underestimate the amount of language contamination in the XGLM pre-training data.

## B Statistical Tests

We provide the full results of the statistical tests for XGLM 4.5B (Table 2), the PolyLMs (Table 3), and the remaining XGLMs (Table 4).

---

[2]https://data.statmt.org/cc-100/

[3]See https://github.com/google/cld3/blob/master/src/nnet_language_identifier.h and https://github.com/google/cld3/blob/master/src/nnet_language_identifier.cc.

| Language Model | Study | Language Pair | F | df$_1$ | df$_2$ | p |
|---|---|---|---|---|---|---|
| XGLM 4.5B | Bernolet et al. (2013) | Dutch→English_Target | 151.98 | 1 | 144 | <0.0001 |
| | Bernolet et al. (2013) | English→Dutch_Target | 24.00 | 1 | 141 | <0.0001 |
| | Cai et al. (2012) Experiment 1/2 | Mandarin→Mandarin_Target | 192.37 | 1 | 24 | <0.0001 |
| | Cai et al. (2012) Experiment 3 | Mandarin→Mandarin_Target | 419.66 | 1 | 32 | <0.0001 |
| | Fleischer et al. (2012) | English→Polish_Target | 1.35 | 1 | 31 | 0.2955 |
| | Fleischer et al. (2012) | Polish→English_Target | 0.96 | 1 | 32 | 0.3704 |
| | Hartsuiker et al. (2004) | English→Spanish_Target | 9.17 | 1 | 112 | 0.0056 |
| | Hartsuiker et al. (2004) | Spanish→English_Target | 4.33 | 1 | 112 | 0.0558 |
| | Kotzochampou et al. (2022) | English→Greek_Target | 7.28 | 1 | 24 | 0.0201 |
| | Kotzochampou et al. (2022) | Greek→English_Target | 5.05 | 1 | 24 | 0.0485 |
| | Kotzochampou et al. (2022) | Greek→Greek_Target | 8.40 | 1 | 24 | 0.0132 |
| | Loebell et al. (2003) | English→German_Target | 0.13 | 1 | 16 | 0.7462 |
| | Loebell et al. (2003) | German→English_Target | 0.10 | 1 | 16 | 0.7647 |
| | Schoonbaert et al. (2007) | Dutch→Dutch_Target | 385.71 | 1 | 144 | <0.0001 |
| | Schoonbaert et al. (2007) | Dutch→English_Target | 57.28 | 1 | 144 | <0.0001 |
| | Schoonbaert et al. (2007) | English→Dutch_Target | 134.53 | 1 | 137 | <0.0001 |

Table 2: Statistical tests of structural priming for XGLM 4.5B.

| Language Model | Study | Language Pair | F | df$_1$ | df$_2$ | p |
|---|---|---|---|---|---|---|
| PolyLM 1.7B | Bernolet et al. (2013) | Dutch→English_Target | 116.87 | 1 | 144 | <0.0001 |
| | Bernolet et al. (2013) | English→Dutch_Target | 18.80 | 1 | 144 | <0.0001 |
| | Cai et al. (2012) Experiment 1/2 | Mandarin→Mandarin_Target | 164.45 | 1 | 24 | <0.0001 |
| | Cai et al. (2012) Experiment 3 | Mandarin→Mandarin_Target | 228.25 | 1 | 32 | <0.0001 |
| | Fleischer et al. (2012) | English→Polish_Target | 7.50 | 1 | 32 | 0.0165 |
| | Fleischer et al. (2012) | Polish→English_Target | 7.34 | 1 | 32 | 0.0174 |
| | Hartsuiker et al. (2004) | English→Spanish_Target | 2.47 | 1 | 112 | 0.1498 |
| | Hartsuiker et al. (2004) | Spanish→English_Target | 1.76 | 1 | 112 | 0.2280 |
| | Kotzochampou et al. (2022) | English→Greek_Target | 0.13 | 1 | 24 | 0.7462 |
| | Kotzochampou et al. (2022) | Greek→English_Target | 0.13 | 1 | 24 | 0.7462 |
| | Kotzochampou et al. (2022) | Greek→Greek_Target | 8.50 | 1 | 24 | 0.0128 |
| | Loebell et al. (2003) | English→German_Target | 1.39 | 1 | 16 | 0.2955 |
| | Loebell et al. (2003) | German→English_Target | 2.66 | 1 | 16 | 0.1525 |
| | Schoonbaert et al. (2007) | Dutch→Dutch_Target | 105.51 | 1 | 144 | <0.0001 |
| | Schoonbaert et al. (2007) | Dutch→English_Target | 55.84 | 1 | 144 | <0.0001 |
| | Schoonbaert et al. (2007) | English→Dutch_Target | 140.97 | 1 | 144 | <0.0001 |
| PolyLM 13B | Bernolet et al. (2013) | Dutch→English_Target | 193.43 | 1 | 144 | <0.0001 |
| | Bernolet et al. (2013) | English→Dutch_Target | 16.73 | 1 | 144 | 0.0002 |
| | Cai et al. (2012) Experiment 1/2 | Mandarin→Mandarin_Target | 141.67 | 1 | 24 | <0.0001 |
| | Cai et al. (2012) Experiment 3 | Mandarin→Mandarin_Target | 257.28 | 1 | 32 | <0.0001 |
| | Fleischer et al. (2012) | English→Polish_Target | 2.45 | 1 | 32 | 0.1570 |
| | Fleischer et al. (2012) | Polish→English_Target | 0.29 | 1 | 32 | 0.6275 |
| | Hartsuiker et al. (2004) | English→Spanish_Target | 21.87 | 1 | 112 | <0.0001 |
| | Hartsuiker et al. (2004) | Spanish→English_Target | 41.60 | 1 | 112 | <0.0001 |
| | Kotzochampou et al. (2022) | English→Greek_Target | 0.70 | 1 | 24 | 0.4481 |
| | Kotzochampou et al. (2022) | Greek→English_Target | 0.54 | 1 | 24 | 0.5062 |
| | Kotzochampou et al. (2022) | Greek→Greek_Target | 9.03 | 1 | 24 | 0.0106 |
| | Loebell et al. (2003) | English→German_Target | 5.36 | 1 | 16 | 0.0485 |
| | Loebell et al. (2003) | German→English_Target | 1.51 | 1 | 16 | 0.2794 |
| | Schoonbaert et al. (2007) | Dutch→Dutch_Target | 260.25 | 1 | 144 | <0.0001 |
| | Schoonbaert et al. (2007) | Dutch→English_Target | 129.76 | 1 | 144 | <0.0001 |
| | Schoonbaert et al. (2007) | English→Dutch_Target | 58.52 | 1 | 144 | <0.0001 |

Table 3: Statistical tests of structural priming for PolyLM 1.7B and 13B.

| Language Model | Study | Language Pair | F | df₁ | df₂ | p |
|---|---|---|---|---|---|---|
| XGLM 564M | Bernolet et al. (2013) | Dutch→English_Target | 12.89 | 1 | 144 | 0.0010 |
| | Bernolet et al. (2013) | English→Dutch_Target | 16.59 | 1 | 144 | 0.0002 |
| | Cai et al. (2012) Experiment 1/2 | Mandarin→Mandarin_Target | 301.39 | 1 | 24 | <0.0001 |
| | Cai et al. (2012) Experiment 3 | Mandarin→Mandarin_Target | 1006.36 | 1 | 32 | <0.0001 |
| | Fleischer et al. (2012) | English→Polish_Target | 1.05 | 1 | 32 | 0.3497 |
| | Fleischer et al. (2012) | Polish→English_Target | 10.30 | 1 | 32 | 0.0056 |
| | Hartsuiker et al. (2004) | English→Spanish_Target | 0.51 | 1 | 112 | 0.5076 |
| | Hartsuiker et al. (2004) | Spanish→English_Target | 4.72 | 1 | 112 | 0.0471 |
| | Kotzochampou et al. (2022) | English→Greek_Target | 5.90 | 1 | 24 | 0.0352 |
| | Kotzochampou et al. (2022) | Greek→English_Target | 11.25 | 1 | 24 | 0.0051 |
| | Kotzochampou et al. (2022) | Greek→Greek_Target | 10.80 | 1 | 24 | 0.0056 |
| | Loebell et al. (2003) | English→German_Target | 3.65 | 1 | 16 | 0.1001 |
| | Loebell et al. (2003) | German→English_Target | 2.76 | 1 | 16 | 0.1494 |
| | Schoonbaert et al. (2007) | Dutch→Dutch_Target | 545.14 | 1 | 144 | <0.0001 |
| | Schoonbaert et al. (2007) | Dutch→English_Target | 5.66 | 1 | 144 | 0.0291 |
| | Schoonbaert et al. (2007) | English→Dutch_Target | 55.69 | 1 | 144 | <0.0001 |
| XGLM 1.7B | Bernolet et al. (2013) | Dutch→English_Target | 17.64 | 1 | 144 | 0.0001 |
| | Bernolet et al. (2013) | English→Dutch_Target | 32.57 | 1 | 144 | <0.0001 |
| | Cai et al. (2012) Experiment 1/2 | Mandarin→Mandarin_Target | 751.15 | 1 | 24 | <0.0001 |
| | Cai et al. (2012) Experiment 3 | Mandarin→Mandarin_Target | 1519.71 | 1 | 32 | <0.0001 |
| | Fleischer et al. (2012) | English→Polish_Target | 0.08 | 1 | 32 | 0.7761 |
| | Fleischer et al. (2012) | Polish→English_Target | 0.69 | 1 | 32 | 0.4481 |
| | Hartsuiker et al. (2004) | English→Spanish_Target | 4.76 | 1 | 112 | 0.0467 |
| | Hartsuiker et al. (2004) | Spanish→English_Target | 3.19 | 1 | 112 | 0.1026 |
| | Kotzochampou et al. (2022) | English→Greek_Target | 2.62 | 1 | 24 | 0.1502 |
| | Kotzochampou et al. (2022) | Greek→English_Target | 11.20 | 1 | 24 | 0.0051 |
| | Kotzochampou et al. (2022) | Greek→Greek_Target | 18.49 | 1 | 24 | 0.0005 |
| | Loebell et al. (2003) | English→German_Target | 1.80 | 1 | 16 | 0.2358 |
| | Loebell et al. (2003) | German→English_Target | 3.13 | 1 | 16 | 0.1247 |
| | Schoonbaert et al. (2007) | Dutch→Dutch_Target | 312.38 | 1 | 144 | <0.0001 |
| | Schoonbaert et al. (2007) | Dutch→English_Target | 3.72 | 1 | 144 | 0.0770 |
| | Schoonbaert et al. (2007) | English→Dutch_Target | 55.88 | 1 | 134 | <0.0001 |
| XGLM 2.9B | Bernolet et al. (2013) | Dutch→English_Target | 47.12 | 1 | 144 | <0.0001 |
| | Bernolet et al. (2013) | English→Dutch_Target | 27.25 | 1 | 144 | <0.0001 |
| | Cai et al. (2012) Experiment 1/2 | Mandarin→Mandarin_Target | 427.12 | 1 | 24 | <0.0001 |
| | Cai et al. (2012) Experiment 3 | Mandarin→Mandarin_Target | 1363.62 | 1 | 32 | <0.0001 |
| | Fleischer et al. (2012) | English→Polish_Target | 1.31 | 1 | 32 | 0.2988 |
| | Fleischer et al. (2012) | Polish→English_Target | 12.11 | 1 | 32 | 0.0031 |
| | Hartsuiker et al. (2004) | English→Spanish_Target | 4.61 | 1 | 112 | 0.0489 |
| | Hartsuiker et al. (2004) | Spanish→English_Target | 10.42 | 1 | 112 | 0.0033 |
| | Kotzochampou et al. (2022) | English→Greek_Target | 3.58 | 1 | 24 | 0.0966 |
| | Kotzochampou et al. (2022) | Greek→English_Target | 12.26 | 1 | 24 | 0.0036 |
| | Kotzochampou et al. (2022) | Greek→Greek_Target | 16.05 | 1 | 24 | 0.0011 |
| | Loebell et al. (2003) | English→German_Target | 6.22 | 1 | 16 | 0.0362 |
| | Loebell et al. (2003) | German→English_Target | 1.11 | 1 | 16 | 0.3485 |
| | Schoonbaert et al. (2007) | Dutch→Dutch_Target | 327.66 | 1 | 144 | <0.0001 |
| | Schoonbaert et al. (2007) | Dutch→English_Target | 21.01 | 1 | 144 | <0.0001 |
| | Schoonbaert et al. (2007) | English→Dutch_Target | 90.89 | 1 | 144 | <0.0001 |
| XGLM 7.5B | Bernolet et al. (2013) | Dutch→English_Target | 37.88 | 1 | 144 | <0.0001 |
| | Bernolet et al. (2013) | English→Dutch_Target | 21.46 | 1 | 144 | <0.0001 |
| | Cai et al. (2012) Experiment 1/2 | Mandarin→Mandarin_Target | 402.46 | 1 | 24 | <0.0001 |
| | Cai et al. (2012) Experiment 3 | Mandarin→Mandarin_Target | 1193.10 | 1 | 32 | <0.0001 |
| | Fleischer et al. (2012) | English→Polish_Target | 0.08 | 1 | 32 | 0.7761 |
| | Fleischer et al. (2012) | Polish→English_Target | 8.96 | 1 | 32 | 0.0093 |
| | Hartsuiker et al. (2004) | English→Spanish_Target | 16.41 | 1 | 112 | 0.0002 |
| | Hartsuiker et al. (2004) | Spanish→English_Target | 17.28 | 1 | 112 | 0.0002 |
| | Kotzochampou et al. (2022) | English→Greek_Target | 3.10 | 1 | 24 | 0.1202 |
| | Kotzochampou et al. (2022) | Greek→English_Target | 12.33 | 1 | 24 | 0.0036 |
| | Kotzochampou et al. (2022) | Greek→Greek_Target | 9.47 | 1 | 24 | 0.0092 |
| | Loebell et al. (2003) | English→German_Target | 1.86 | 1 | 16 | 0.2310 |
| | Loebell et al. (2003) | German→English_Target | 6.84 | 1 | 16 | 0.0291 |
| | Schoonbaert et al. (2007) | Dutch→Dutch_Target | 402.81 | 1 | 144 | <0.0001 |
| | Schoonbaert et al. (2007) | Dutch→English_Target | 43.84 | 1 | 144 | <0.0001 |
| | Schoonbaert et al. (2007) | English→Dutch_Target | 83.09 | 1 | 144 | <0.0001 |

Table 4: Statistical tests of structural priming for XGLM 564M, 1.7B, 2.9B, and 7.5B.