# OpenReview forum: "Structural Priming Demonstrates Abstract Grammatical Representations in Multilingual Language Models"
_EMNLP/2023/Conference — EMNLP 2023 Main_

### Official Review · Reviewer_ACY8 · 2023-07-28

**Soundness:** 4

**Excitement:**

4: Strong: This paper deepens the understanding of some phenomenon or lowers the barriers to an existing research direction.

**Missing References:**

Some typological investigations into shared, abstract language representations in MLMs could be added, for example:
- *Investigating Language Relationships in Multilingual Sentence Encoders Through the Lens of Linguistic Typology* -- Choenni & Shutova (2022)

**Paper Topic And Main Contributions:**

The paper investigates *structural priming* in the context of multi-lingual language models. Structural priming is an important paradigm within psycholinguistics, which over the past 4 decades has been one of the main tools for uncovering the presence of abstract, linguistic structure within human language processing.

Recently, structural priming has been deployed as a tool for uncovering linguistic structure within language model representations. The authors build on this work, and introduce a cross-lingual experimental setup in which a language model is primed in 1 language, and tested in another language. The authors pose that this setup is able to uncover abstract structure even better than in a monolingual setting, since the caveat of token overlap between prime- and target sentences is no longer present.

The authors leverage a wide range of cross-lingual experiments from prior psycholinguistic work, and find that the MLM they consider (XGLM) generally follows similar patterns as have been reported for humans.

**Questions For The Authors:**

It is not entirely clear to me how the Priming Effect is being computed. You state around L197 that in psycholinguistic studies researchers often investigate the proportion of sentences of a particular structure being **produced**. The distinction between production and comprehension is important within priming, and when transferring these paradigms to LLMs it is important to be clear about what is being investigated.

In your case you are not letting the models generate a target open-endedly, but you only compare the probabilities of the two structural alterations. I suggest you address this more explicitly; having a concrete example of your setup in Section 3.2 would improve the clarity.

------

Contrary to Sinclair et al. (2022), you compare model probabilities of two different *target* sentences. In Sinclair et al., the authors purposely opted to only alternate the *prime* sentences, and compare the probability of the same target sentence. This ensures that length differences between alternating targets play role, or prior probabilities needing to be corrected for (as in Prasad et al. (2019), see also Fig. 1 in Sinclair et al. (2022)). From the paragraph around L224 it is not entirely clear to me if your mixed-effects model fully addresses these concerns: could these factors play a role in your results?

------

Should I interpret your priming effect measure as 0.5 having no significant impact at all? For example, both PO/DO yielding the same probability, as explained in the paragraph around L197?

**Reasons To Accept:**

- The paper provides a highly interesting addition to the priming paradigm within LLM interpretability research, with the introduction of a multilingual approach that has played a major role within psycholinguistics research.
- The paper engages deeply with the existing psycholinguistics literature on the subject, which results in a rich and broad experimental setup tackling a wide range of languages.

**Reasons To Reject:**

- The procedure for measuring priming effect is not explained in enough detail, and could raise some concerns (see my Question below).

**Reproducibility:**

5: Could easily reproduce the results.

**Reviewer Confidence:**

5: Positive that my evaluation is correct. I read the paper very carefully and I am very familiar with related work.

**Typos Grammar Style And Presentation Improvements:**

Very well written!

As stated in one of my questions in the field above, having a concrete example of a model + prime/target + model probabilities + final priming effect would help a lot with interpreting the experimental results.

---

> ### Author Rebuttal · Authors · 2023-08-29
>
> We thank the reviewer for their feedback. We address specific comments below:
>
> *"It is not entirely clear to me how the Priming Effect is being computed. You state around L197 that in psycholinguistic studies researchers often investigate the proportion of sentences of a particular structure being produced. The distinction between production and comprehension is important within priming, and when transferring these paradigms to LLMs it is important to be clear about what is being investigated.
> In your case you are not letting the models generate a target open-endedly, but you only compare the probabilities of the two structural alterations. I suggest you address this more explicitly; having a concrete example of your setup in Section 3.2 would improve the clarity."*
>
> *"As stated in one of my questions in the field above, having a concrete example of a model + prime/target + model probabilities + final priming effect would help a lot with interpreting the experimental results."*
>
> We thank the reviewer for noting that the exact way in which the priming effect is calculated is not clearly described in the submitted version of the manuscript, and agree that including this will help to clear up some of the methodological details and results. We will include a concrete example of how the priming effect is calculated in the updated manuscript. A sketch of this is given below:
>
> Consider the following two primes and the following two targets:
>
> DO prime: The nun shows the prisoner a hat.
> PO prime: The nun shows a hat to the prisoner.
> DO target: The chef gives the swimmer a hat.
> PO target: The chef gives a hat to the swimmer.
>
> We can then use language models to calculate the probability of each target following each prime. As an example, the language models can provide probabilities as follows:
>
> P(PO Target | DO Prime) = 0.30
> P(DO Target | DO Prime) = 0.20
> P(PO Target | PO Prime) = 0.40
> P(DO Target | PO Prime) = 0.10
>
> We then normalize the probabilities of each target given its prime (i.e. restrict the probabilities to the two possible targets). In the submitted version of the paper, these normalized probabilities were referred to as 'relative probabilities'. In the final version of the paper, we will change 'relative probabilities' to 'normalized probabilities'.
>
> Normalized P(PO Target | DO Prime) = 0.60
> Normalized P(DO Target | DO Prime) = 0.40
> Normalized P(PO Target | PO Prime) = 0.80
> Normalized P(DO Target | PO Prime) = 0.20
>
> Because the probabilities of the two targets following a given prime now sum to one, we only need to consider the probabilities for one target type (e.g. PO Targets). We then run our analyses by comparing the normalized probabilities for that target type (similar to Sinclair (2022)). For example, if we were to compare the relative probabilities of PO Targets, we would compare the following two probabilities:
>
> Normalized P(PO Target | DO Prime) = 0.60
> Normalized P(PO Target | PO Prime) = 0.80
>
> In this case, we see a clear structural priming effect—the presence of the PO prime increases the relative probability of a PO Target by 0.20. Our statistical models take multiple such instances to see whether this is a general trend across stimuli. Because they are symmetric, we only calculate the priming effect for one target type (e.g. the PO Target above), following the lead of the original study in determining which target type to use to calculate the effect. It is the normalized probabilities of these targets that are plotted in the graphs in the paper.
>
> *"Contrary to Sinclair et al. (2022), you compare model probabilities of two different target sentences. In Sinclair et al., the authors purposely opted to only alternate the prime sentences, and compare the probability of the same target sentence. This ensures that length differences between alternating targets play role, or prior probabilities needing to be corrected for (as in Prasad et al. (2019), see also Fig. 1 in Sinclair et al. (2022)). From the paragraph around L224 it is not entirely clear to me if your mixed-effects model fully addresses these concerns: could these factors play a role in your results?"*
>
> It is indeed true that the prior probabilities might be biased towards one structure (e.g. the PO target is always more probable than the DO target in the example above, regardless of prime type). We hope that the description above has clarified how we account for this—like Sinclair et al. (2022), our priming effect is based on the *difference* between the probability of a target given each prime. The main difference is that to make our results more comparable with many structural priming studies where the proportions of each target response are compared, we use the difference between the normalized probabilities to operationalize the effect rather than the difference between the un-normalized log-transformed probabilities.
>
> *"Should I interpret your priming effect measure as 0.5 having no significant impact at all? For example, both PO/DO yielding the same probability, as explained in the paragraph around L197?"*
>
> We hope that the previous description has helped to make our priming effect measure clearer. To clarify: if a PO and DO target have the same probability given a specific prime (e.g., a PO prime), then their normalized probability will each be 0.5. In this case, the priming effect will be calculated as the difference between this and the normalized probability of the same target given a DO prime. This is different from a priming effect of 0.5, which would mean a 0.5 difference between the normalized probability of a target given each prime (e.g. the difference between the normalized probability of the PO target given a PO prime and the normalized probability of the PO target given a DO prime).
>
> *"Some typological investigations into shared, abstract language representations in MLMs could be added, for example:
> Investigating Language Relationships in Multilingual Sentence Encoders Through the Lens of Linguistic Typology -- Choenni & Shutova (2022)"*
>
> We thank the reviewer for bringing this paper to our attention. We will include a brief discussion about possible effects of linguistic typology on shared abstract grammatical representations in the revised manuscript.

---

### Official Review · Reviewer_1oDq · 2023-08-04

**Typos Grammar Style And Presentation Improvements:** 228
**Soundness:** 3

**Excitement:**

3: Ambivalent: It has merits (e.g., it reports state-of-the-art results, the idea is nice), but there are key weaknesses (e.g., it describes incremental work), and it can significantly benefit from another round of revision. However, I won't object to accepting it if my co-reviewers champion it.

**Paper Topic And Main Contributions:**

The authors apply methods from human structural priming studies to language models (LMs) and find that models that are primed with a grammatical structure favor (in terms of probability) utterances with the same structure. They demonstrate this effect even when the priming example and target utterance are in different languages. The authors present this as evidence that these models have abstract grammatical representations.

**Reasons To Accept:**

This paper has a compelling experimental design, and generally I am excited about comparing results from human linguistic experiments to results from the same experiments on LMs.

**Reasons To Reject:**

My concern with this paper is whether the effect on probability given the intervention is sufficient evidence to conclude that LMs have abstract grammatical representations (AGRs). The majority of the experiments where the effects or structural priming are seen are between relatively closely related languages, where similar grammatical structures might result in surface-level similarities, like word order, which may account for the effects seen in the experiments. The phenomenon discussed in the paragraphs at L509 and L529 would seem to support this hypothesis. In my view, this paper would be stronger if there was more discussion of alternative hypotheses to AGRs that would explain their results and perhaps reasons to reject them.

**Reproducibility:**

5: Could easily reproduce the results.

**Reviewer Confidence:**

3: Pretty sure, but there's a chance I missed something. Although I have a good feel for this area in general, I did not carefully check the paper's details, e.g., the math, experimental design, or novelty.

---

> ### Author Rebuttal · Authors · 2023-08-29
>
> We thank the reviewer for their feedback. Specific comments are addressed below:
>
> *"My concern with this paper is whether the effect on probability given the intervention is sufficient evidence to conclude that LMs have abstract grammatical representations (AGRs). The majority of the experiments where the effects or structural priming are seen are between relatively closely related languages, where similar grammatical structures might result in surface-level similarities, like word order, which may account for the effects seen in the experiments. The phenomenon discussed in the paragraphs at L509 and L529 would seem to support this hypothesis. In my view, this paper would be stronger if there was more discussion of alternative hypotheses to AGRs that would explain their results and perhaps reasons to reject them."*
>
> We thank the reviewer for noting this concern, and we agree that a revision should address alternate hypotheses and the nuanced interpretation of results more directly. We note that while many of the language pairs studied are related, there is no overlap in the words or morphemes used to construct each structure across any pair of languages. From this we can infer that even if a given word order primes a homologous word order in a related language, this still demonstrates abstraction at least at the level of word order. Whether word order can be considered to be a form of abstract grammatical representation is hotly debated. For instance, under theories of the cognitive representation of language such as that proposed by Branigan and Pickering (2017), abstract grammatical representations are precisely defined as those  structures that can be structurally primed (such as specific word orders). Other theories disagree. We will clarify the implications of the study's findings for different definitions of abstract grammatical representations.
>
> *"'We include a random intercept for experimental item' does not make sense to me."*
>
> By this we mean that the linear mixed-effects regression includes a random intercept for each experimental item. This accounts for effects of individual experimental items, independent of prime condition.
>
> *"'DO (double object)' should be dative I believe."*
>
> It is indeed true that the DO is a dative, but in this case we are distinguishing between two specific kinds of dative: the double object (DO) dative and the prepositional object (PO) dative.
>
> *"Sections 4.{1,2}.\* should have been a table in my opinion. It reads as highly repetitive."*
>
> We thank the reviewer for this suggestion, and will consider this for the next version of the paper.

---

### Official Review · Reviewer_HSdV · 2023-08-06

**Soundness:** 3

**Excitement:**

3: Ambivalent: It has merits (e.g., it reports state-of-the-art results, the idea is nice), but there are key weaknesses (e.g., it describes incremental work), and it can significantly benefit from another round of revision. However, I won't object to accepting it if my co-reviewers champion it.

**Paper Topic And Main Contributions:**

The main contribution of this paper is to demonstrate via crosslingual structural priming that multilingual language models have abstract linguistic representations within and across languages.

**Questions For The Authors:**

- Can you elaborate on the advantage of (crosslingual) structural priming over various forms of probing to investigate abstract linguistic representations of neural language models?
- Have you compared monolingual and multilingual language models against monolingual structural priming paradigms to test whether multilingual training reinforces abstract linguistic representations of multilingual language models?

**Reasons To Accept:**

- This paper comprehensively investigates abstract linguistic representations of multilingual language models via structural priming based on 12 (8 crosslingual + 4 monolingual) structural priming paradigms in total.

**Reasons To Reject:**

- This paper seems to over-interpret the results as "reinforc[ing] the claims of usage-based (i.e. functional) accounts of language acquisition".

**Reproducibility:**

5: Could easily reproduce the results.

**Reviewer Confidence:**

4: Quite sure. I tried to check the important points carefully. It's unlikely, though conceivable, that I missed something that should affect my ratings.

---

> ### Author Rebuttal · Authors · 2023-08-29
>
> We thank the reviewer for their feedback. We address specific comments below:
>
> *"This paper seems to over-interpret the results as 'reinforc[ing] the claims of usage-based (i.e. functional) accounts of language acquisition'."*
>
> We thank the reviewer for alerting us to the possible overly-broad interpretation of this statement. By mentioning that the results 'reinforce the claims of usage-based (i.e. functional) accounts of language acquisition', we simply mean that they demonstrate the in-principle viability of learning the kinds of abstract linguistic representations that are sensitive to structural priming using the statistics of language alone. We will clarify this in the revised version of the paper.
>
> *"Can you elaborate on the advantage of (crosslingual) structural priming over various forms of probing to investigate abstract linguistic representations of neural language models?"*
>
> As noted only briefly on L130-142, one limitation of probing studies is that they often do not demonstrate causal effects of representations on model outputs. Even if a model represents similar structures across languages similarly, the model may not necessarily use the representations in functionally the same way across languages. In some cases, the representational similarity may be something learned by the probe but not the models themselves (Belinkov, 2022: Probing Classifiers: Promises, Shortcomings, and Advances). Multilingual structural priming demonstrates higher-order abstractions across languages with causal effects on model outputs, and it does not require access to internal model states or any assumptions about model architecture.
>
> In addition, one limitation of using monolingual structural priming to assess language models is that because humans demonstrate structural priming, language models may learn associations between specific words such that they appear to show effects of structural priming without learning any abstract patterns. By using crosslinguistic structural priming, we mitigate this effect for language models that are not trained on large amounts of data involving code-switching. Discussion of this point will also be included in the revised draft.
>
> *"Have you compared monolingual and multilingual language models against monolingual structural priming paradigms to test whether multilingual training reinforces abstract linguistic representations of multilingual language models?"*
>
> We agree that this is an interesting question. However, because current pretrained monolingual and multilingual models are not trained on the same datasets (or even on datasets of the same size), and because there is evidence that ostensibly 'monolingual' pretrained language models may often be exposed to large amounts of data in other languages (Blevins and Zettlemoyer, 2022: Language Contamination Helps Explains the Cross-lingual Capabilities of English Pretrained Models), this would require training our own models to run these analyses, and thus is beyond the scope of the present study.

---

### Meta-Review · Area_Chair_K5Pg · 2023-09-17

**Recommendation:** 4

**Metareview:**

The paper applies structural priming, a method in psycholinguistics for studying the presence of linguistic structure in human language processing, to multilingual language models. A cross-lingual setup for priming is followed, testing multiple grammatical alterations. The results show that language model behaviour correlates with those of human subjects. The reviewers expressed some concerns about an overly broad interpretation of the results, however the study still makes a valuable contribution to the question studied.

---

### Decision · Program_Chairs · 2023-10-07

**Decision:**

Accept-Main

**Comment:**

The paper applies structural priming, a method in psycholinguistics for studying the presence of linguistic structure in human language processing, to multilingual language models. A cross-lingual setup for priming is followed, testing multiple grammatical alterations. The results show that language model behaviour correlates with those of human subjects. The reviewers expressed some concerns about an overly broad interpretation of the results, however the study still makes a valuable contribution to the question studied.